# Decision Mamba: A Multi-Grained State Space Model with Self-Evolution Regularization for Offline RL

**Qi Lv**[1,2]  **Xiang Deng**[1,†]  **Gongwei Chen**[1]  **Michael Yu Wang**[2]  **Liqiang Nie**[1,†]

[1]School of Computer Science and Technology, Harbin Institute of Technology (Shenzhen)
[2]School of Engineering, Great Bay University
`lvqi@stu.hit.edu.cn`

## Abstract

While the conditional sequence modeling with the transformer architecture has demonstrated its effectiveness in dealing with offline reinforcement learning (RL) tasks, it is struggle to handle out-of-distribution states and actions. Existing work attempts to address this issue by data augmentation with the learned policy or adding extra constraints with the value-based RL algorithm. However, these studies still fail to overcome the following challenges: (1) insufficiently utilizing the historical temporal information among inter-steps, (2) overlooking the local intra-step relationships among return-to-gos (RTGs), states and actions, (3) overfitting suboptimal trajectories with noisy labels. To address these challenges, we propose **D**ecision **M**amba (**DM**), a novel multi-grained state space model (SSM) with a self-evolving policy learning strategy. DM explicitly models the historical hidden state to extract the temporal information by using the mamba architecture. To capture the relationship among RTG-state-action triplets, a fine-grained SSM module is designed and integrated into the original coarse-grained SSM in mamba, resulting in a novel mamba architecture tailored for offline RL. Finally, to mitigate the overfitting issue on noisy trajectories, a self-evolving policy is proposed by using progressive regularization. The policy evolves by using its own past knowledge to refine the suboptimal actions, thus enhancing its robustness on noisy demonstrations. Extensive experiments on various tasks show that DM outperforms other baselines substantially.

## 1 Introduction

Offline Reinforcement Learning (RL) [13, 27, 29, 38] has attracted great attention due to its remarkable successes in the fields of robotic control [5, 36] and games [3, 32, 50]. As transformer [49] has exhibited powerful sequential modeling abilities in natural language processing [4, 43] and computer vision [10, 42], many efforts [6, 8, 25, 61] have been made on applying this architecture to offline RL tasks. Transformer-based methods view the reward/return-to-go (RTG), state, and action as a sequence, and then predict actions by using the transformer encoder. However, it often fails to make correct decisions when encountering out-of-distribution states or actions, showing limited robustness. Previous work attempts to address this issue from the perspective of data augmentation [51, 64] and objective constraints [6, 53, 61]. However, they introduce a significant number of noises or the overestimation bias. Thus, how to enhance model robustness remains a highly challenging and insufficiently explored issue.

In this study, we offer two novel perspectives on improving model robustness through both the model architecture and learning strategy. In terms of the model architecture, (1) although previous

---

[†]Corresponding Author.

38th Conference on Neural Information Processing Systems (NeurIPS 2024).

studies have made some modifications to the transformer architecture [23, 44, 52], they have not fully utilized inter-step information, particularly historical information which is critical for decision-making processes. For example, the robot can adjust its subsequent routes based on the historical information of failed paths for completing the navigation task; (2) furthermore, most existing approaches adopt transformer to model the flattened trajectory as a sequence, while ignoring the structural trajectory patterns of the causal intra-step relationship among **R**TGs, **s**tates, and **a**ctions (RSAs). A RL policy typically predicts the next action given the current state based on the RTG. Thus, this kind of fine-grained intrinsic connection among RSAs is intuitively beneficial for policy learning. As regards to the learning strategy, (3) there exists a large number of noisy labels in the suboptimal trajectories which hurt the performance of the policy significantly. Although the existing work that generates pseudo trajectories or actions alleviates this problem to some extent [57, 64], it also introduces other biases or errors.

To address the above issues, we propose **D**ecision **M**amba (**DM**), a multi-grained state space model with a self-evolving policy learning strategy for offline RL. In order to adequately leverage the historical information, we adopt mamba to explicitly model the temporal state among inter-steps, since mamba architecture [15, 18, 40] shows a more effective capability of extracting the historical information. Meanwhile, the causal intra-step relationship is beneficial for the model to understand the common patterns within the local dynamics. Thus, we introduce a fine-grained SSM module to extract the local features of structural patterns among the RSA triplet within each intra-step. Apart from modifying the model architecture and aligning it to the trajectory pattern, we also propose a learning strategy to prevent the policy from overfitting noisy labels. This is achieved by a progressive self-evolution regularization which leverages the past knowledge of the policy itself to refine and adjust the target label adaptively.

We conduct comprehensive experiments on Gym-Mujoco and Antmaze benchmark, containing 5 tasks with varying levels of noise and difficulties. The performance of DM surpasses other baselines by approximately 8% with respect to the average normalized score on the three classic Mujoco tasks, showing its effectiveness. In summary, the contributions of this paper are summarized as follows:

- Different from the existing conditional sequence modeling work for offline RL with the transformer architecture, we propose Decision Mamba (DM), a generic offline RL backbone built on State Space Models, which leverages the historical temporal information sufficiently for robust decision making.

- To extract the casual intra-step relationships, we introduce a fine-grained SSM module and integrate it to the original coarse-grained SSM in mamba, which combines the local trajectory patterns with the global sequential features, achieving the multi-grained modeling capability.

- To prevent the policy from overfitting the noise trajectories, we adopt a self-evolving policy learning strategy to progressively refine the target, which uses the past knowledge of the learned policy itself as an additional regularizer to constrain the training objective.

## 2 Related Work

### 2.1 Offline Reinforcement Learning with Transformer-based Models

Offline Reinforcement Learning (RL) [7, 13, 22, 27, 29, 38, 56, 57, 67] is widely used for robotic control and decision-making. In particular, transformer-based methods [8, 25, 44] reformulate the trajectories as a state/action/RTG sequence, and predict the next action based on the historical trajectories. However, although the sequence modeling methods formulate offline RL in a simplified form, they can hardly deal with the overfitting problem caused by the suboptimial trajectories in offline data [11, 21, 59]. One line of approaches [34, 51, 64, 66] focused on exploiting data augmentation methods, such as generating additional data via the bootstrap method, or training an inverse dynamics model to predict actions for the large amount of unlabelled trajectories. Another line of work [6, 23, 25, 35, 44, 52] attempted to modify the transformer architecture to explicitly make use of the structural patterns within the training data. Furthermore, substantial efforts [37, 54, 58, 62] have also been made on applying regularization terms to learning policies, such as RvS [11] and QDT [61]. Nevertheless, previous work simply applies transformer to offline RL tasks while seldom considering about adapting the architecture to trajectory learning. Thus, these methods fail to extract

the historical information sufficiently and are unable to capture local patterns thoroughly from the trajectories. In this work, we address these issues by proposing DM, a tailored mamba architecture for offline RL tasks. A fine-grained SSM module is designed in DM to supply fine-grained intra-step information to the coarse-grained inter-steps features. Together with the architecture, we also present a self-evolving policy learning strategy to prevent the model from overfitting noise labels.

## 2.2 State Space Models for Linear-time Sequence Modeling

Recently, State Space Models (SSMs) show high potentials in various domains, including natural language processing [15–17, 19, 20, 40, 46], computer vision [30, 31, 33, 41, 63, 65] and time-series forecasting [55]. Stemming from signal processing, SSMs capture global dependencies from a sequence more effective in a lightweight structure and shows advantages in compressing the historical information, compared with the transformer architecture. Although SSMs have considerable benefits, it still struggles to perform contextual reasoning. Mamba [15] is thus proposed to alleviate this problem. It introduced a time-varying selective mechanism and a hardware-friendly design, making it as a competitive architecture against with transformer. The Mamba architecture is then adapted to different downstream tasks by considering the characteristics of these tasks. VIM [65] and VMamba [33] introduced 2D SSMs for image understanding. VideoMamba [30] introduced spatio-temporal scan for video understanding. In this work, we take the fine-grained trajectory patterns into consideration, and introduce a multi-grained mamba architecture tailored for RL tasks.

# 3 Method

## 3.1 Preliminaries

**Decision Transformer for Offline RL.** The fundamental Markov Decision Process [12] can be represented as $\mathcal{M} = (\mathcal{S}, \mathcal{A}, \mathcal{T}, r, \gamma)$, where $\mathcal{S}$ is the state space, $\mathcal{A}$ is the action space, $\mathcal{T} : \mathcal{S} \times \mathcal{A} \to \mathcal{S}$ is the transition function, $r : \mathcal{S} \times \mathcal{A} \to \mathbb{R}$ is the reward function, and $\gamma \in (0, 1]$ is the discount factor. Given an offline dataset $D_\mu$ collected by the behavior policy $\mu(a|s)$, offline RL algorithms aim to maximize the rewards. Formally, the iteration process of learning a policy is as below ($k$ denotes the index of the learning iteration):

$$Q_k^\pi = \underset{Q}{\arg\min} \, \mathbb{E}_{(s,a,r,s') \sim D_\mu}[Q(s,a) - (r + \gamma \mathbb{E}_{a' \sim \pi_{k-1}(\cdot|s')} Q_{k-1}^\pi(s', a'))]^2, \tag{1}$$

$$\pi_k = \underset{\pi}{\arg\max} \, \mathbb{E}_{s \sim D_\mu}[\mathbb{E}_{a \sim \pi(\cdot|s)} Q_k^\pi(s, a)] \ \text{ s.t. } \mathbb{E}_{s \in D_\mu}[D(\pi(\cdot|s), \mu(\cdot|s))] \le \epsilon. \tag{2}$$

When updating the Q function, $(s, a, r, s')$ are sampled from $D_\mu$ but the target action $a'$ is sampled from the current policy $\pi_{k-1}$.

Inspired by the great success of sequence generation models in NLP [9, 39, 48], Decision Transformer [8] is proposed to model the trajectory optimization problem as an action prediction procedure. Specifically, it first obtains the return-to-go (RTG) with the reward, i.e., $R_t = \sum_{i=t}^{T} r_i$. Then, the learned policy, which is based on the decoder-only transformer architecture [48], predicts the action sequence $a_i$ autoregressively, with the offline trajectory $\tau = (s_0, R_0, a_0, \dots, s_T, R_T, a_T)$. The training objective is as follows:

$$\underset{\theta}{\text{minimize}} \ \mathcal{J}(\pi_\theta^k) = \mathbb{E}_\tau \Big[ \sum_{t=1}^{T} -\log \pi_\theta^k(a_t | \tau_{t-l:t}) \Big] \tag{3}$$

where $\tau_{t-l:t} = (s_j, R_j, a_j, \dots, s_t, R_t) \ (j = \min(t - l, 0))$ is the input trajectory and $l$ is the length of context window.

**SSMs for Linear-Time Sequence Modeling.** The State Space Model (SSM) describes the probabilistic dependence between the continuous input signal $x(t)$ and the observed output $y(t)$ via the latent hidden state $h(t)$ as Eq. (5):

$$h'(t) = \boldsymbol{A}h(t) + \boldsymbol{B}x(t), \tag{4}$$

$$y(t) = \boldsymbol{C}h(t). \tag{5}$$

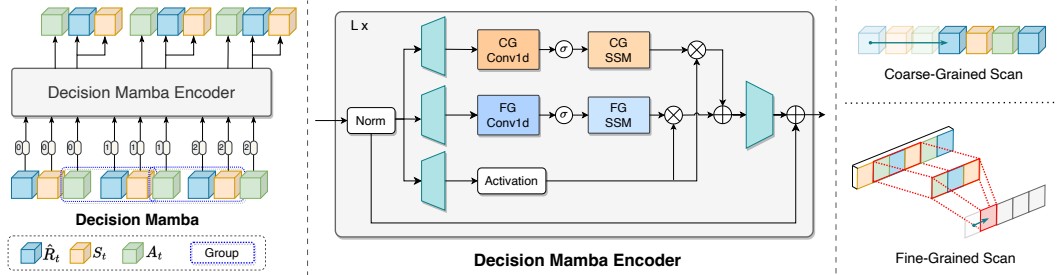

Figure 1: Model Overview. *The left*: we combine the trajectories $\tau$ with position embeddings, and then feed the result sequence to the Decision Mamba encoder which has $L$ layers. *The middle*: a coarse-grained branch and a fine-grained branch are integrated together to capture the trajectory features. *The right*: visualization of multi-grained scans.

In order to apply the SSM model to the discrete input sequence instead of the original continuous signal, Structured SSM (S4) [18] discretizes it by a step size $\Delta$ as Eq. (7).

$$h_t = \overline{\boldsymbol{A}}h_{t-1} + \overline{\boldsymbol{B}}x_t, \tag{6}$$

$$y_t = \boldsymbol{C}h_t, \tag{7}$$

where $\overline{\boldsymbol{A}} = \exp(\Delta\boldsymbol{A})$, $\overline{\boldsymbol{B}} = (\Delta\boldsymbol{A})^{-1}(\exp(\Delta\boldsymbol{A}) - \boldsymbol{I}) \cdot \Delta\boldsymbol{B}$. To this end, the model can forward-propagate in an efficient parallelizable mode with a global convolution. Due to the linear time invariance brought by the SSM model, it lacks the content-aware reasoning ability which is important in sequence modeling. Therefore, mamba [15] proposes the selective SSM which adds the length dimension to the original parameters $(\Delta, \boldsymbol{B}, \boldsymbol{C})$, changing it from time-invariant to time-varying. It uses a parameterized projection to project the size of parameter $\boldsymbol{B}, \boldsymbol{C}$ from $(D, N)$ to $(B, L, N)$, and $\Delta$ from $(D)$ to $(B, L, D)$, where $D, B, L$ and $N$ denotes the channel size, batch size, sequence length and hidden size, respectively.

### 3.2 Decision Mamba

The transformer architecture has been well used in offline RL tasks. Despite its strong ability to understand complete trajectory sequence, it shows limited capabilities in capturing historical information. Thus, we propose a multi-grained space state model to extract the fine-grained local information to supply the coarse-grained global information, namely Decision Mamba (DM), for comprehensively learning the trajectory representation. Figure 1 presents the overall framework of DM.

#### 3.2.1 Multi-Grained Mamba

**Trajectory Embeddings**  Following the sequence modeling [8], we first use multilayer perceptrons (MLPs) to embed the RSAs from the given trajectory $\tau = (R_0, s_0, a_0, \ldots, R_T, s_T, a_T)$. Then, the trajectory embeddings are added the absolute step position embeddings to attach the position information, similar to the classic usage in the NLP field. Mathematically, it can be formulated as follows:

$$e_i^{\mathrm{R}} = \mathrm{MLP}(R_i), \quad e_i^{\mathrm{s}} = \mathrm{MLP}(s_i), \quad e_i^{\mathrm{a}} = \mathrm{MLP}(a_i), \tag{8}$$

$$e_i = [e_i^{\mathrm{R}}; e_i^{\mathrm{s}}; e_i^{\mathrm{a}}] + \mathrm{broadcast}(e_i^{\mathrm{t}}), \tag{9}$$

where $e_i^{\mathrm{f}} \in \mathbb{R}^{B \times L \times N}$, and $[;]$ denotes the concatenate operation.

**Coarse-Grained SSM**  Different from the transformer-based methods [8, 25], DM models the historical information before the current $i$-th step via the latent hidden state $h_i$ as shown in Eq. (6). It explicitly represents the feature of history information, rather than only learns such information implicitly. As the number of encoder layers increases, historical information is selectively preserved in the representation $h_i$. To this end, DM is expected to have a better capability to understand the sequential dependencies. It can be formulated as follows:

$$h_i^{\mathrm{CG}} = \mathrm{SiLU}(\mathrm{Proj}(h_i)); \quad h_i^{\mathrm{CG}} = \mathrm{InterS3M}(h_i^{\mathrm{CG}}), \tag{10}$$

where $h_i^{\mathrm{CG}}$ represents the coarse-grained hidden state; $\mathrm{InterS3M}$ denotes the coarse-grained SSM.

**Fine-Grained SSM** Further, to better discern the dependencies among RSA within each step, we gather the feature of each single step to obtain the fine-grained representation via a 1D-convolution layer, and then introduce a fine-grained SSM module for extracting the local pattern among RSA, as shown in the middle part and right part of Figure 1.

It can be formulated as follows:

$$h_i^{\text{FG}} = \text{Conv1D}(h_i), \tag{11}$$

$$h_i^{\text{FG}} = \text{SiLU}(\text{Proj}(h_i^{\text{FG}})), \tag{12}$$

$$h_i^{\text{FG}} = \text{IntraS3M}(h_i^{\text{FG}}), \tag{13}$$

where $h_i^{\text{FG}}$ indicates the fine-grained hidden state; IntraS3M means the fine-grained SSM.

**Fusion Module** For gathering both fine-grained local trajectory patterns and coarse-grained global contextual information, we combine the $h_i^{\text{FG}}$ with $h_i^{\text{CG}}$ in each encoder layer and then use the layer normalization to ensure that the multi-grained features have a consistent distribution. In order to remain the important historical information, we add a residual connection. The fusion process can be formulated as follows:

$$h_i^{\text{MG}} = \text{LN}(h_i^{\text{CG}} + h_i^{\text{FG}}), \tag{14}$$

$$h_i = \text{Proj}(h_i^{\text{MG}} + h_{i-1}), \tag{15}$$

---

**Algorithm 1** Decision Mamba

**Input:** trajectory sequence $\tau = (R_0, s_0, a_0, \ldots, R_t, s_t)$
**Output:** action $a_t$
1: /* obtain the embedding of trajectory sequence*/
2: $\mathbf{R}, \mathbf{s}, \mathbf{a}$: (B, L) $\leftarrow$ Split($\tau_{t-l:t}$)
3: $\mathbf{e^R}, \mathbf{e^s}, \mathbf{e^a}$: (B, L, D) $\leftarrow$ MLP($\mathbf{R}$), MLP($\mathbf{s}$), MLP($\mathbf{a}$)
4: $\mathbf{h}_0$: (B, L, D) $\leftarrow$ Flatten($\mathbf{e^R}, \mathbf{e^s}, \mathbf{e^a}$)
5: **for** $i$ in layer **do**
6: $\quad \mathbf{h}_i^{\text{CG}}$ : (B, L, D) $\leftarrow$ Norm($\mathbf{h}_{i-1}$)
7: $\quad \mathbf{h}_i^{\text{FG}}$ : (B, L, D) $\leftarrow$ Conv1d$_i^{\text{CG}}$($\mathbf{h}_i^{\text{CG}}$)
8: $\quad \mathbf{z}^{\text{CG}}$ : (B, L, D) $\leftarrow$ Linear$_i^{\text{CG}}$($\mathbf{h}_{i-1}$)
9: $\quad \mathbf{z}^{\text{FG}}$ : (B, L, D) $\leftarrow$ Linear$_i^{\text{FG}}$($\mathbf{h}_{i-1}$)
10: $\quad$ /* process with multi-grained branchs */
11: $\quad$ **for** $\mathbf{h}^f$ in ($\mathbf{h}_i^{\text{CG}}, \mathbf{h}_i^{\text{FG}}$) **do**
12: $\qquad \mathbf{h}_i^f$ : (B, L, D) $\leftarrow$ SiLU(Conv1d$_i^f$($\mathbf{h}^f$))
13: $\qquad \mathbf{A}_i^f$ : (D, N) $\leftarrow$ Parameter
14: $\qquad \mathbf{B}_i^f$ : (B, L, N) $\leftarrow$ Linear$_i^{f\mathbf{B}}$($\mathbf{h}_i^f$)
15: $\qquad \mathbf{C}_i^f$ : (B, L, N) $\leftarrow$ Linear$_i^{f\mathbf{C}}$($\mathbf{h}_i^f$)
16: $\qquad \boldsymbol{\Delta}_i^f$ : (B, L, D) $\leftarrow$ log($1 + \exp($Linear$_i^{f\boldsymbol{\Delta}}$($\mathbf{h}_i^f$) + Parameter$_i^{f\boldsymbol{\Delta}}$))
17: $\qquad \overline{\mathbf{A}}_i^f, \overline{\mathbf{B}}_i^f$: (B, L, D, N) $\leftarrow$ discretize($\boldsymbol{\Delta}_i^f \mathbf{A}_i^f, \mathbf{B}_i^f$)
18: $\qquad \mathbf{h}_i^f$ : (B, L, D) $\leftarrow$ SSM($\overline{\mathbf{A}}_i^f, \overline{\mathbf{B}}_i^f, \mathbf{C}_i^f$)($\mathbf{h}_i^f$)
19: $\quad$ **end for**
20: $\quad \mathbf{h}_i^{\text{CG}}$ : (B, L, D) $\leftarrow$ $\mathbf{h}^{\text{CG}} \odot$ SiLU($\mathbf{z}$)
21: $\quad \mathbf{h}_i^{\text{FG}}$ : (B, L, D) $\leftarrow$ $\mathbf{h}^{\text{FG}} \odot$ SiLU($\mathbf{z}$)
22: $\quad$ /* fusion of multi-grained features */
23: $\quad \mathbf{h}_i^{\text{MG}}$ : (B, L, D) $\leftarrow$ LayerNorm($\mathbf{h}_i^{\text{CG}} + \mathbf{h}_i^{\text{FG}}$)
24: $\quad \mathbf{h}_i$ : (B, L, D) $\leftarrow$ Linear($\mathbf{h}_i^{\text{MG}} + \mathbf{h}_{i-1}$)
25: **end for**
26: $a_t$ : (B, L) $\leftarrow$ MLP($\mathbf{h}$)
27: **return** $a_t$

---

where $h_i^{\text{MG}}$ indicates the multi-grained hidden state and LN denotes the layer normalization. The forward propagation procedure of DM is presented in Algorithm 1.

### 3.2.2 Progressive Self-Evolution Regularization

There are typically amounts of suboptimal trajectories in RL tasks. The previous approaches usually overfit these noisy data and thus lack robustness. Fortunately, the existing literature [26] has shown that deep models learn clean samples (optimal trajectories) at the beginning of the training process, and then overfit the noisy samples (suboptimal trajectories). Inspired by this observation, we propose a *progressive self-evolution regularization* (PSER), which uses the knowledge of the past policy to refine the noisy labels as supervision for policy learning, thus avoiding fitting the noisy trajectories.

Figure 2: The process of PSER includes: i) generating action labels with previous step policy, ii) refining target label, iii) computing loss, where the red circle denotes the noise.

Specifically, we obtain a refined target by combining the ground truth and the prediction from the learned policy itself. Let $\hat{a}_k$ denote the prediction about $s$ from the current policy $\pi_k(a|s)$ at $k$-th iteration. The refined target at $k$-th can be written as follows:

$$\tilde{a}_k = (1 - \beta) \, a_k + \beta \, \hat{a}_{k-1}, \tag{16}$$

where $\hat{a}_{k-1} \sim \pi_{k-1}(\cdot|s)$ and $\beta$ is the trade-off weight.

To obtain more insights about the refined targets Eq. (16), we compare the gradients of the training objectives with the original label and the refined label. The standard Mean Square Error (MSE) loss function of Eq. (3) with the original label can be written as:

$$\mathcal{L}_k(\hat{a}_k, a_k) = ||\hat{a}_k - a_k||^2. \tag{17}$$

In comparison, the loss function with the refined target of Eq. (16) can be rewritten as:

$$\mathcal{L}_{SE,k}(\hat{a}_k, a_k) = ||\hat{a}_k - \tilde{a}_k||^2 = ||\hat{a}_k - (1 - \beta)\, a_k - \beta\, \hat{a}_{k-1}||^2. \tag{18}$$

Comparing the objectives of Eq. (17) and Eq. (18), the gradient of $\mathcal{L}_{SE,k}$ with respect to the output of policy $\{a_{k,i}\}_{i=1}^T$ can be derived by:

$$\frac{\partial \mathcal{L}_{SE,k}}{\partial a_{k,i}} = 2\left[\underbrace{(\hat{a}_{k,i} - a_{k,i})}_{\nabla \mathcal{L}_k} - \beta\, \underbrace{(\hat{a}_{k-1,i} - a_{k,i})}_{\nabla \mathcal{R}}\right], \tag{19}$$

where $\nabla \mathcal{L}_k$ indicates the gradient of the loss function Eq. (17), and $\nabla \mathcal{R}$ computes the difference between the past predictions and the targets. From the perspective of gradient back propagation, PSER imposes a regularization constraint on the current policy $\pi_k(a|s)$ by smoothing the original target action $a_{k,i}$ with the self-generated label $\hat{a}_{k-1,i}$.

Moreover, it is important to determine the value of $\beta$ in Eq. (18). The $\beta$ controls the learning procedure, where the policy trusts the given actions if $\beta$ is set to a large value. As stated above, the policy tends to fit gradually from clean patterns to noisy patterns. Thus, we set the $\beta$ to dynamically increased values. As $\beta$ increases, the policy progressively gains more confidence in its own past knowledge. To maintain the learning process stable, we apply the linear growth approach and set a lower boundary. The $\beta$ at the $k$-th iteration is computed as follows:

$$\beta_k = \max(\beta_K \times \frac{k}{K}, \beta_{\min}), \tag{20}$$

where $K$ is the number of total iterations for training and $\beta_K$ is the hyperparameter.

We replace the original label with the refined target, leading to the objective:

$$\underset{\theta}{\text{minimize}}\ \mathbb{E}_{s_t, \tau \sim D_\mu}\left[\log \pi_\theta(\tilde{a}_t | R_t, s_t, \tau_{<t})\right]. \tag{21}$$

We adopt the MSE loss, and then the objective 21 is converted to:

$$\mathcal{L}_{\text{PSE},k}(\hat{a}_k, a_k) = ||\hat{a}_k - \tilde{a}_k||^2 = ||\hat{a}_k - (1 - \beta_k)\, a_k - \beta_k\, \hat{a}_{k-1}||^2. \tag{22}$$

### 3.2.3 Training Objective

To make the training procedure more robust, we introduce the inverse training goals: predicting the next state and the next RTG. Individuals often assess the feasibility of actions by envisioning their potential outcomes. Therefore, we expect the policy to predict the post-execution state and RTG based on the predicted action, thus improving its robustness. Specifically, given the trajectory $\tau = (R_0, s_0, a_0, \ldots, R_t, s_t)$, Decision Mamba originally predicts the next action $\hat{a}_t$. Further, by incorporating the action $a_t$ to the original trajectory $\tau_{t-l:t}$, it is also predicts the next RTG $\hat{R}_{t+1}$ and the next state $\hat{s}_{t+1}$. Compared to the Eq. (3), the training objective of DM with the refined target can be written as follows:

$$\underset{\theta}{\text{minimize}}\ \mathbb{E}_{s_t, \tau \sim D_\mu}\left[\sum_{t=0}^T \left[\lambda_1 \underbrace{\log \pi_\theta(\tilde{a}_t | R_t, s_t, \tau_{<t})}_{\text{PSER}} + \lambda_2 \underbrace{\log \pi_\theta(R_{t+1} | \tau_{\leq t})}_{\text{predicting RTGs}} + \lambda_3 \underbrace{\log \pi_\theta(s_{t+1} | \tau_{\leq t})}_{\text{predicting states}}\right]\right], \tag{23}$$

where the $\tilde{a}_t$ is computed by Eq. (16), $\lambda_i$ is the weight hyperparameter, and the sum of $\lambda_i$ is set to 1. Note, we omit the length of context window $l$ for simplicity.

## 4 Experiment

### 4.1 Settings

**Dataset and Evaluation Metrics.** We conduct our experiments on *Gym-MuJoCo* which is one of the mainstream benchmarks used in offline deep RL [14, 28, 59], including Hopper, HalfCheetah, Walker and Ant tasks. Each task contains medium, medium-expert, medium-replay and expert datasets. To more comprehensively evaluate our proposed method, we also adopt the *AntMaze* benchmark which is a navigation task of aiming to reach a fixed goal location, with the 8-DoF "Ant" quadraped robot. We evaluate Decision Mamba by using the popular suite D4RL [13]. Following the existing literature [8, 25], we normalize the score for each dataset roughly for comparison, by computing normalized score $= 100 \times \frac{\text{score} - \text{random score}}{\text{expert score} - \text{random score}}$. More details about dataset and implementation can be found in Appendix A.

**Baselines.** We compare Decision Mamba with existing SOTA offline RL approaches including Behavioral Cloning (BC), Conservative Q-Learning (CQL) [29], Decision Transformer (DT) [8], Reinforcement Learning via Supervised Learning (RvS) [11], StARformer (StAR) [44], Graph Decision Transformer (GDT) [23], Waypoint Transformer (WT) [2], Elastic Decision Transformer (EDT) [60], and Language Models for Motion Control (LaMo) [45]. Among these methods, CQL stands as a representative of value-based methods, while the other methods belong to supervised learning (SL) approaches. For most of these baselines, we cite the results from the original papers. In addition, we reimplement DT and LaMo for more comparison in different settings by using their repositories. The detailed descriptions of these baselines are presented in Appendix B.

| Dataset | BC | CQL$^\dagger$ | DT | RvS$^\dagger$ | StAR$^\dagger$ | GDT$^\dagger$ | WT$^\dagger$ | EDT | LaMo | DM (Ours) |
|---|---|---|---|---|---|---|---|---|---|---|
| HalfCheetah-M | 42.2 | **44.4** | 42.6 | 41.6 | 42.9 | 42.9 | 43.0 | 42.5 | 43.1 | 43.8 $_{\pm 0.23}$ |
| Hopper-M | 55.6 | 86.6 | 70.4 | 60.2 | 65.8 | 77.1 | 63.1 | 63.5 | 74.1 | **98.5** $_{\pm 8.19}$ |
| Walker-M | 71.9 | 74.5 | 74.0 | 73.9 | 77.8 | 76.5 | 74.8 | 72.8 | 73.3 | **80.3** $_{\pm 0.07}$ |
| HalfCheetah-M-E | 41.8 | 62.4 | 87.3 | 92.2 | **93.7** | 93.2 | 93.2 | 48.5 | 92.2 | 93.5 $_{\pm 0.11}$ |
| Hopper-M-E | 86.4 | 110.0 | 106.5 | 101.7 | 110.9 | 111.1 | 110.9 | 110.4 | 109.9 | **111.9** $_{\pm 1.84}$ |
| Walker-M-E | 80.2 | 98.7 | 109.2 | 106.0 | 109.3 | 107.7 | 109.6 | 108.4 | 108.8 | **111.6** $_{\pm 3.31}$ |
| HalfCheetah-M-R | 2.2 | **46.2** | 37.4 | 38.0 | 39.9 | 40.5 | 39.7 | 37.8 | 39.5 | 40.8 $_{\pm 0.43}$ |
| Hopper-M-R | 23.0 | 48.6 | 82.7 | 82.2 | 81.6 | 85.3 | 88.9 | 89.0 | 82.5 | **89.1** $_{\pm 4.32}$ |
| Walker-M-R | 47.0 | 32.6 | 66.6 | 66.2 | 74.8 | 77.5 | 67.9 | 74.8 | 76.7 | **79.3** $_{\pm 1.94}$ |
| **Avg.** | 50.0 | 67.1 | 75.8 | 71.7 | 77.4 | 79.1 | 78.7 | 72.0 | 77.8 | **83.2** $_{\pm 0.82}$ |

Table 1: Overall Performance. M, M-E, and M-R denotes the medium, medium-expert, and medium-replay, respectively. The results of the baselines marked with $^\dagger$ are cited from their original papers. We report the mean and standard deviation of the normalized score with four random seeds. **Bold** and underline indicate the highest score and second-highest score, respectively.

## 4.2 Overall Results

For a fair comparison, we first conduct experiments on datasets commonly adopted by mainstream approaches. The overall performance is presented in Table 1. It can be observed that Decision Mamba outperforms other baselines in most datasets. On one hand, benefiting from the supervised learning objective, SL-based baselines exhibit a strong ability in the high-quality datasets (M-E), but show weakness in the suboptimal datasets (M/M-R). On the other hand, CQL performs well in the suboptimal datasets due to regularizing the Q-values during training, but struggles to perform well in the high-quality datasets.

For DM, it shows significant improvement over the other SL-based methods. Specifically, it outperforms the best of the baselines by 4%+, especially in suboptimal datasets, e.g., on the medium datasets, the performance of DM surpasses the value-based method CQL and the transformer-based method GDT by around 6% and 9% on average, respectively. This significant improvement demonstrates the robustness of DM in learning from suboptimal datasets, attributed to the multi-grained mamba encoder and PSER module in DM. Note, although DM performs slightly worse than CQL on the Halfcheetah-M-R and HalfCheetah-M datasets, the difference is not significant. Therefore, the

| Dataset | BC | DT | LaMo | DM (Ours) |
|---|---|---|---|---|
| HalfCheetah-E | 83.3 | 90.5 | 92.0 | **93.5** $_{\pm 0.23}$ |
| Hopper-E | 90.2 | 109.6 | 111.6 | **112.5** $_{\pm 0.75}$ |
| Walker-E | 103.2 | 108.1 | 108.1 | **108.3** $_{\pm 0.13}$ |
| Ant-M | 91.0 | 95.3 | 94.6 | **104.8** $_{\pm 1.40}$ |
| Ant-M-E | 99.8 | 129.6 | 134.8 | **136.2** $_{\pm 0.36}$ |
| Ant-M-R | 79.5 | 81.4 | **92.7** | 89.5 $_{\pm 1.64}$ |
| Ant-E | 112.6 | 123.1 | 134.2 | **135.9** $_{\pm 0.35}$ |
| Antmaze-U | 63.0 | 63.0 | 80.0 | **100.0** $_{\pm 0.08}$ |
| Antmaze-U-D | 61.0 | 61.0 | 70.0 | **90.0** $_{\pm 0.10}$ |
| **Avg.** | 87.1 | 95.7 | 102.0 | **107.9** $_{\pm 3.33}$ |

Table 2: Extensive Results. E, U, and U-D denotes the expert, umazed, and umazed-diverse.

proposed DM shows stronger overall performance, capable of learning from both high-quality and suboptimal datasets simultaneously.

In order to evaluate our method more comprehensively, we also conduct experiments on other datasets. We adopt representative baselines including BC, DT and LaMo, where LaMo leverages extensive additional natural language corpora and knowledge to enhance the model performance. As illustrated in Table 2, all methods perform exceptionally well on the Expert dataset. However, when it comes to mixing the suboptimal data into the training set, compared with DT, both LaMo and DM exhibit

significant superiority, while DM shows a more pronounced overall enhancement. For instance, DM outperforms LaMo by approximately 10% on the Ant-M dataset and around 6% on average. For *AntMaze*, it requires composing parts of suboptimal trajectories to form more optimal policies for reaching goals. "U-D" is more difficult than "U", and DM shows superiority in these tasks. More comparison results can be found in Appendix C.

## 4.3 Ablation Study

To investigate the effectiveness of each component in DM, we conduct experiments with different variants of DM. In particular, we compare 3 different implementations: (1) *w/o MG* removes the multi-grained branch, directly using the sequence feature original extracted from mamba; (2) *w/o PSER* removes the progressive self-evolution regularization, training the model with the labels in the training dataset; (3) *w/o ILO* removes the inverse learning objective, only predicting the action in the training procedure. As shown in Table 3, the performance of DM drops significantly without either of these components. Notably, the most substantial performance degradation with about 6% occurs when the PSER module is removed, especially in the suboptimal datasets. This observation verifies the effectiveness of this module in preventing policy from overfitting and thus enhancing its robustness. MG and ILO are also critical for offline RL tasks. Once these two modules are excluded, there is a noticeable reduction in the model's performance.

|  | Halfcheetah | | | Hopper | | | Walker | | | Avg. |
|---|---|---|---|---|---|---|---|---|---|---|
|  | M | M-E | M-R | M | M-E | M-R | M | M-E | M-R | |
| **DM** | **43.8** | **93.5** | **40.8** | **98.5** | **111.9** | **89.1** | **80.3** | **111.6** | **79.3** | **83.2** |
| *w/o MG* | 43.3 | 92.9 | 40.1 | 86.2 | 111.2 | 77.5 | 79.2 | 107.9 | 74.9 | 79.2 |
| *w/o PSER* | 42.9 | 91.0 | 37.5 | 85.3 | 110.4 | 76.4 | 76.2 | 105.6 | 69.6 | 77.2 |
| *w/o ILO* | 43.1 | 92.3 | 39.4 | 94.0 | 110.7 | 85.3 | 80.1 | 108.8 | 73.5 | 80.8 |

Table 3: Ablation Results. "*w/o MG/PSER/ILO*" represents removing the module of multi-grained feature extraction, the progressive self-evolution regularization, and inverse learning objectives, respectively. **Best** results are marked in bold.

## 4.4 Comparison Results with Different Context Lengths

To validate whether DM can capture the information of inter-step and intra-step, we investigate the performance of our model with different context lengths. We conduct experiments with the context length $L = \{20, 40, 60, 80, 100, 120\}$. Figure 3 shows the comparison results. Regardless of different context lengths, the proposed DM consistently achieves a high score than other baselines among all datasets, showcasing its superiority in capturing the inter-step dependencies in different lengths.

It is noteworthy that BC shows a comparable performance to DT in the Hopper-M and Halfcheetah-M datasets, yet demonstrates a substantial discrepancy in other datasets. We deduce that, in addition to the inherent limitations of the imitation learning paradigm, the BC model that relies solely on MLP also has significant architectural disadvantages. Consequently, it can achieve scores of only 60-70% at most on the expert datasets. When trained on M-R data, BC evidently struggles to learn effectively, achieving only approximately 30% and less than 4% performance on Hopper-M-R and Halfcheetah-M-R, respectively. Due to the attention and SSM mechanisms, DT and DM models conspicuously exhibit a higher upper bound compared to BC. Among them, our proposed DM shows the best performance across all datasets. This indicates that the specific architecture of DM enables it to extract more useful information from the inter-step and intra-step, leading to a strong performance across different context lengths.

## 4.5 The Effects of $\beta$ in PSER

We have shown that the proposed PSER in DM enhances the robustness of the policy significantly in learning on suboptimal trajectories. Consequently, we endeavor to delve deeper into the impact of the policy self-evolution throughout the training process. During the training procedure, $\beta_K$ in PSER determines the upper bound of the policy self-evolution. When $\beta_K$ is set to 1, the policy has the highest dependency on self-learned knowledge; conversely, if $\beta_K$ is set to 0, the policy tends to completely lose its ability to self-evolve. We conduct experiments on DM variants, by removing the lower boundary $\beta_{\min}$ and selecting $\beta_K$ from the set $\{1, 0.75, 0.5, 0.25, 0\}$. The results are depicted in

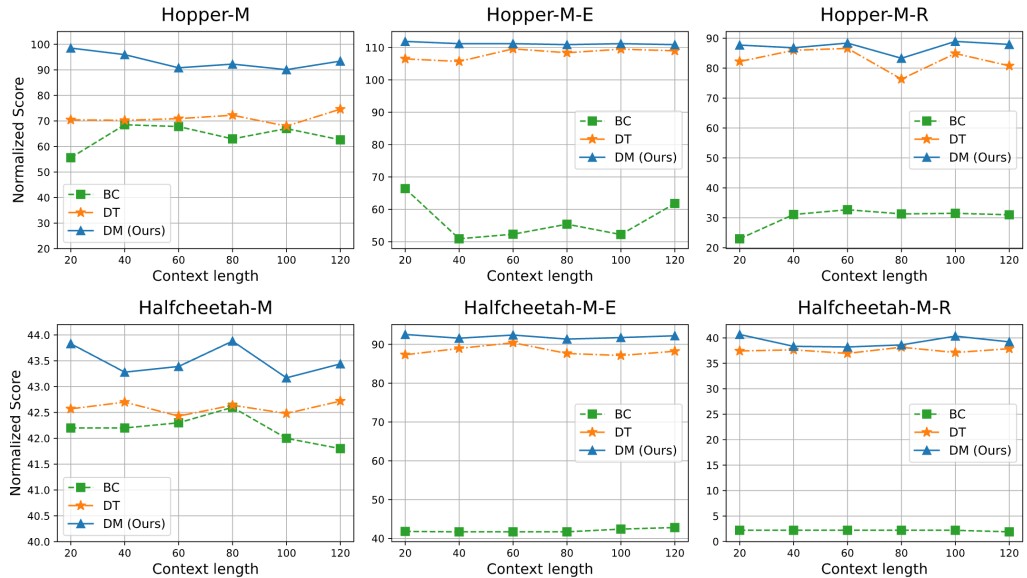

Figure 3: Impact of Context Lengths. We compare the normalized scores of BC, DT and DM with different context lengths. The DM consistently outperforms other baselines.

Table 4. It can be observed that if we remove the self-evolution capability of DM, i.e., setting $\beta_K$ to 0, there is a notable decline in the performance across both the M and M-R correlated datasets, with the poorest overall performance. When increasing the $\beta$ to 0.5, the model obtains the best performance since it reaches a good balance between the ground truth and the learned knowledge. To prevent the policy from excessively relying on its past knowledge, DM additionally uses a lower boundary $\beta_{\min}$ (set to 0.5), achieving the best performance.

| | Halfcheetah | | | Hopper | | | Walker | | | Avg. |
|---|---|---|---|---|---|---|---|---|---|---|
| | M | M-E | M-R | M | M-E | M-R | M | M-E | M-R | |
| DM ($\beta_K = 1$) | 43.5 | 92.3 | 38.4 | 97.9 | 111.3 | 82.7 | 77.8 | 109.4 | 74.7 | 80.9 |
| DM ($\beta_K = 0.75$) | 42.8 | 91.9 | 38.7 | 98.4 | 110.5 | 83.8 | 77.6 | 106.2 | 71.3 | 80.3 |
| DM ($\beta_K = 0.5$) | **43.9** | 92.1 | 38.8 | **98.6** | 111.1 | 86.6 | 77.2 | 108.6 | 75.8 | 81.4 |
| DM ($\beta_K = 0.25$) | 43.8 | 91.5 | 38.6 | 97.7 | 107.0 | 86.4 | 76.9 | 106.2 | 72.8 | 80.2 |
| DM ($\beta_K = 0$) | 42.9 | 91.0 | 37.5 | 85.3 | 110.4 | 76.4 | 76.2 | 105.6 | 69.6 | 77.2 |
| DM | 43.8 | **93.5** | **40.8** | 98.5 | **111.9** | **89.1** | **80.3** | **111.6** | **79.3** | **83.2** |

Table 4: The effects of $\beta$ in PSER.

## 5 Conclusion

In this paper, we study the offline reinforcement learning from the perspectives of the architecture and the learning strategy. We have accordingly proposed Decision Mamba (DM), a multi-grained state space model tailored for RL tasks with a self-evolving policy learning strategy. DM enhances the policy robustness by adapting the mamba architecture to RL tasks by capturing the fine-grained and coarse-grained information. Meanwhile, the proposed learning strategy prevents the policy from overfitting the noisy labels with a progressive self-evolution regularization. Extensive experiments demonstrate that DM outperforms other baselines by approximately 4% on the mainstream offline RL benchmarks, showing its robustness and effectiveness.

**Limitations and Future Directions**. According to [15, 18], the mamba structure is more friendly to long sequences than the transformer structure, not only in terms of capturing historical information, but also in terms of the computational speed. Benefiting from the structure of SSM, the computational complexity of mamba is $O(n)$, while the computational complexity of attention score in transformer is $O(n^2)$. Thus, the computational efficiency of mamba is higher. Although the exploration of computational efficiency is an exciting direction for future research, it is not within the main scope of this paper.

## Acknowledgments and Disclosure of Funding

We would like to thank the reviewers for their constructive comments. This work is supported by Shenzhen College Stability Support Plan (Grant No.GXWD202208817144428005) and National Natural Science Foundation of China (Grant No. 62236003). Additionally, it is also supported by National Natural Science Foundation of China (Grant No. 62406092), and partially supported by Research on Efficient Exploration and Self-Evolution of APP Agents & Embodied Intelligent Cerebellum Control Model and Collaborative Feedback Training Project (Grant No. TC20240403047).

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

## A Dataset and Implementation Details

### A.1 Dataset Details.

We conduct experiments on five tasks of *Mujoco* and *Antmaze* [47] including Halfcheetah, Hopper, Walker, Ant and Antmaze, as illustrated in Figure 4. Note, all datasets we used is the $v2$ version. In these tasks, there are totally 5 different datasets which are described below:

- Medium: A "medium" policy is trained by using the Soft Actor-Critic [21] with early-stopping the training, and generate 1 million timesteps, achieving about one-third the score of an expert policy.

- Medium-Expert: 1 million timesteps generated by the medium policy concatenated with 1 million timesteps generated by an expert policy (a fine-tuned RL policy).

- Medium-Replay: It involves recording all samples in the replay buffer observed during training until the policy achieves a "medium" level of performance.

- Umaze: It contains the trajectories where the ant to reach a specific goal from a fixed start location.

- Umaze-diverse: Different from Umaze, it is a more difficult dataset where the start position is also random.

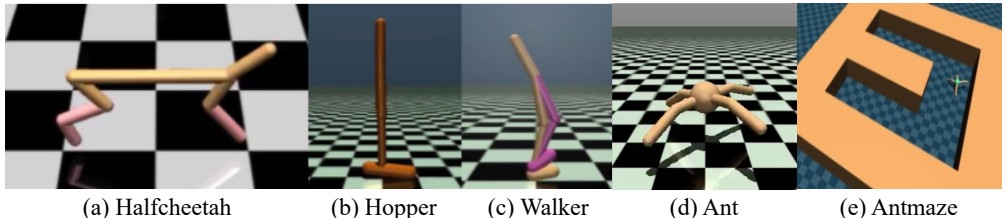

|         (a) Halfcheetah         (b) Hopper        (c) Walker        (d) Ant        (e) Antmaze |

Figure 4: The visualizations of tasks.

### A.2 Implement Details

For all our experiments, we utilized the default hyperparameter settings and conducted 100,000 training iterations or gradient steps. We implement our method with the official repository of the huggingface.The shared hyperparameters are set to the same as those of LaMo, including the batch size, learning rate, overall training steps, and weight decay. The setting of other specific hyperparameters, including $\beta_K$, $\beta_{\min}$ are presented in Table 5. The experiments are conducted on an 8*4090-24G platform, and we run each experiment with four different seeds to ensure its reliability.

| Dataset | Learning Rate | Weight Decay | Context Length | Return-to-go | Training Steps | $\beta_K$ | $\beta_{\min}$ |
|---|---|---|---|---|---|---|---|
| Halfcheetah | $1 \times 10^{-4}$ | $1 \times 10^{-5}$ | 20 | 1800, 3600 | 100K | 0.85 | 0.5 |
| Hopper | $1 \times 10^{-4}$ | $1 \times 10^{-5}$ | 20 | 8000, 12000 | 100K | 0.90 | 0.5 |
| Walker | $1 \times 10^{-4}$ | $1 \times 10^{-5}$ | 20 | 2500, 5000 | 100K | 0.95 | 0.5 |
| Ant | $1 \times 10^{-4}$ | $1 \times 10^{-5}$ | 20 | 3600, 6000 | 100K | 0.85 | 0.5 |
| Antmaze | $1 \times 10^{-4}$ | $1 \times 10^{-5}$ | 20 | 5, 20 | 100K | 0.95 | 0.5 |

Table 5: Task-specific Hyperparameters.

### A.3 Code base

The code bases employed for our evaluations are detailed below.

- BC: https://github.com/kzl/decision-transformer
- DT: https://github.com/kzl/decision-transformer
- EDT: https://github.com/kristery/Elastic-DT
- LaMo: https://github.com/srzer/LaMo-2023

# B  Details of Baselines

We compare our proposed Decision Mamba with previous strong baselines as follows:

- Behavioral Cloning (BC): it is a representative method of imitation learning. The states and actions are collected as the training data first. Then the agent uses a classifier or regressor to replicate the trajectory when encountering the same state.

- Conservative Q-Learning (CQL) [29]: it encourages policies that are less likely to choose actions with high Q-value estimates that are uncertain or unreliable, thus expecting to address overestimation bias.

- Decision Transformer (DT) [8]: it flats the trajectory sequence and use conditional sequence modeling method to autoregressively predict actions.

- RvS [11]: it uses the goal or reward as the condition to realize the behavior cloning. We use the reward-conditioned BC as comparison.

- StARTransformer (StAR) [44]: it extracts the image state patches by self-attending mechanism, then combining the features with the whole sequence.

- Graph Decision Transformer (GDT) [23]: it adopts the sequence modeling method, and models the input sequence into a causal graph to capture relationships among states, actions, and return-to-gos.

- WaypointTransformer (WT) [2]:it integrates intermediate targets and proxy rewards as guidance to steer a policy to desirable outcomes.

- Elastic Decision Transformer (EDT) [60]: it estimates the highest achievable value given a certain history, and inputs the traversed trajectory with a variable length to learn the stitching trajectories.

- Language Models for Motion Control (LaMo) [45]: it adopts the pretrained GPT2 [43] model as the backbone, and use the additional NLP corpus to co-training the policy via the parameter-efficiently LoRA method.

# C  More Comparison

For extensive comparison, we compare DM with more baselines, including the diffusion-based model: Diffuser [24], Decision Diffuser (DD) [1])and more complex approaches: Trajectory Transformer (TT) [25], Critic-Guided Decision Transformer (CGDT) [53]).

As illustrated in Table 6, it can be observed DM still has the strongest overall performance, although it did not achieve the best results on some datasets. Diffusion-based models synthesize optimal trajectories from a generative perspective, showing a significant advantage on replay datasets. Although CGDT performs well, it requires complex additional training, namely Critic training, which increase convergence difficulty. Overall, DM shows superiority on average.

| Dataset | TT$^\dagger$ | CGDT$^\dagger$ | Diffuser$^\dagger$ | DD$^\dagger$ | DM (Ours) |
|---|---|---|---|---|---|
| HalfCheetah-M | **46.9** | 43.0 | 44.2 | 49.1 | 43.8 $_{\pm 0.23}$ |
| Hopper-M | 61.1 | 96.9 | 58.5 | 79.3 | **98.5** $_{\pm 8.19}$ |
| Walker-M | 79.0 | 79.1 | 79.7 | **82.5** | 80.3 $_{\pm 0.07}$ |
| HalfCheetah-M-E | 95.0 | **93.6** | 79.8 | 90.6 | 93.5 $_{\pm 0.11}$ |
| Hopper-M-E | 110.0 | 107.6 | 107.2 | 111.8 | **111.9** $_{\pm 1.84}$ |
| Walker-M-E | 101.9 | 109.3 | 108.4 | 108.8 | **111.6** $_{\pm 3.31}$ |
| HalfCheetah-M-R | 41.9 | 40.4 | **42.2** | 39.3 | 40.8 $_{\pm 0.43}$ |
| Hopper-M-R | 91.5 | 93.4 | 96.8 | **100.0** | 89.1 $_{\pm 4.32}$ |
| Walker-M-R | **82.6** | 78.1 | 61.2 | 75.0 | 79.3 $_{\pm 1.94}$ |
| **Avg.** | 78.9 | 82.4 | 75.3 | 81.8 | **83.2** $_{\pm 0.82}$ |

Table 6: More comparison with other baselines. The results are all cited from their original papers.

## D  The Result on the Distribution of Returns

We compare the ability of policy to understand return-to-go tokens by varying the desired target return over a wide range, especially in the out-of-distribution range. As illustrated in Figure 5, we can observed in the seen target, i.e., on the left side of the yellow dashed line, the expected target returns and the true observed returns are highly correlated. However, when it comes to the out-of-distribution target, the score of DM is consistently higher than those of DT. Among them, due to the extreme difficulty of the HalfCheetah dataset, the performance of DM under the OOD target is only slightly surpassing DT. Conversely, on the other two datasets, DM exhibits strong robustness to the OOD target, significantly outperforming DT. The experimental result has illustrated DM has a strong robustness.

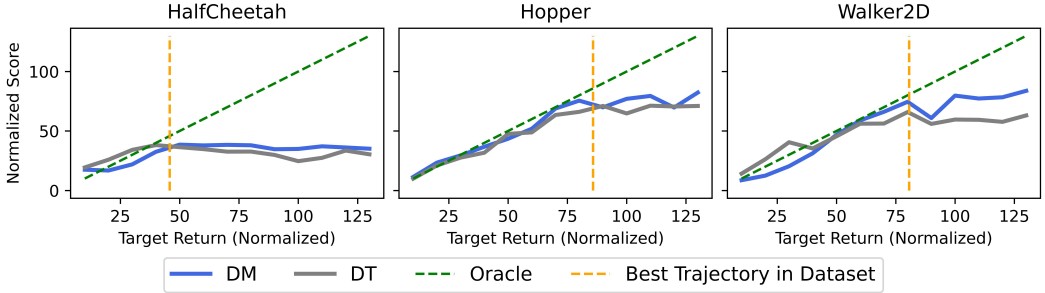

Figure 5: The normalized scores of DT and DM when conditioned on the specified target returns.

## E  Visualization of Action Distribution

We visualize the action distribution of learned policy. Specifically, we use the policies trained on different level of noisy data to predict the next action of the same trajectory, and visualize the hidden layer of the predicted action. As shown in Figure 6, the distribution obtained by DM is more concentrated. This indicates that even if the noise level in the training data varies, DM can still learn an approximate distribution, demonstrating its strong robustness.

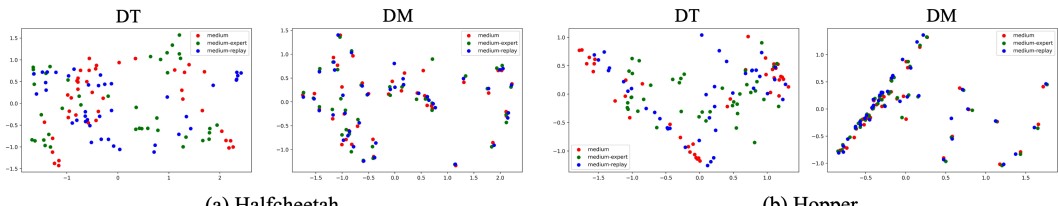

Figure 6: The distributions of action.

## F  Impact/Safegard Statements

**Impact Statement**    In this study, we propose DM, which effectively extracts historical information, fuses multi-grained information to predict action, and enhances the effectiveness of conditional sequence modeling for offline RL tasks. In addition, we introduce a self-evolving policy learning strategy to effectively prevent the policy from overfitting the noisy trajectories, and further enhance the robustness of the policy. To this end, this technology is expected to advance the offline RL agent which can assist human beings in working under the dangerous circumstances. There are many potential societal consequences of developing advanced RL algorithms, none which we feel must be specifically highlighted here.

**Safegrad Statement**    In the paper, we have taken rigorous steps to ensure the responsible release of our new offline RL algorithm and any associated models or data. Given the potential for misuse or

dual-use of such technology, we have implemented several safeguards to mitigate these risks. Use of our algorithms is subject to the CC BY-NC-SA 4.0 agreement. In addition, we are committed to ongoing monitoring and evaluation of our models' usage to identify any emerging risks or patterns of misuse. We will take prompt action if we detect any unauthorized or inappropriate use of our technology. Finally, we are open to working with regulators, researchers, and industry partners to further refine our safeguards and ensure the safe and ethical use of our offline RL algorithm.

