# OpenReview forum: "Decision Mamba: A Multi-Grained State Space Model with Self-Evolution Regularization for Offline RL"
_NeurIPS.cc/2024/Conference — NeurIPS 2024 poster_

### Official Review · Reviewer_qjYZ · 2024-06-20

**Soundness:** 3
**Presentation:** 3
**Contribution:** 3
**Rating:** 6
**Confidence:** 2

**Summary:**

This paper introduces a novel application of state space model (SSM) into the offline RL problem. Prior to this, transformer-based architectures were used heavily, but the authors claim that they did not process “historical information,” which is often an important requirement in real world scenarios. In this work, a Decision Mamba encoder is composed of three stages. First, it embeds its trajectory using three types of MLPs along with time embeddings. Second, coarse-grained SSM directly uses hidden states from a structured SSM for understand sequential dependency. Thirdly, fine-grained use an 1D-convolutional layer for encompassing past few embeddings to precisely understand context. For training, this work proposes a progressive self-evolution regularization (PSER). In experiments, DM archives very promising scores in various offline RL MuJoCo tasks.

**Strengths:**

SSMs are a new class of models that is reported to be successful for various temporal problems; this reviewer believe they need to be tested for various subfields of ML that deal with sequential tasks (presumably equipped with the Markov assumption).  Since SSMs always produce linear-time sequential representation of their states, I think it was reasonable for authors to consider this characteristic as an advantage to solve decision problems. Notably, this paper have following strengths
- This is a good application of SSM into the offline RL tasks.
- The proposed architecture was straightforward to understand
- The experimental results are evaluted in various environments and the proposed model achieved solid numbers.

**Weaknesses:**

- I find the lack of detailed reasoning of the manuscript why Conv1D module brings “fine-grained” control.
- From my perspective, the proposed PSER is related to PPO and DPO methods in terms of regularization. It would be great there was similar theoretical justification (formally or informally) for this proposed loss term.

**Questions:**

I do not see tendency in Fig. 3. Is there results with very few context lengths? eg. {1, 5, 10}

**Limitations:**

The authors authors adequately addressed the limitations.

---

> ### Author Rebuttal · Authors · 2024-08-07
>
> We thank reviewer qjYZ for the valuable comments. We now address the concerns raised in the review below.
>
> > **Q1**: I find the lack of detailed reasoning of the manuscript why Conv1D module brings “fine-grained” control.
>
> **A1**: Intra-step relationships among the **S**tate, **A**ction, and **R**eturn-to-go (**SAR**) is one kind of the fine-grained information. It is well known that convolution neural networks (CNNs) are good at extracting local fine-grained features via small filters and sliding windows. Thus,  we use Conv1D to model the  intra-step dependencies among SAR in each step. The extracted fine-grained features are then fused with the original coarse-grained features in each Decision Mamba block, as shown in Figure 1 in the submission. We have also conducted experiments to validate the effectiveness of fine-grained information. As shown in Table 3 in the submission,  the performance of DM drops significantly without extracting fine-grained features (i.e., DM w/o GB), which demonstrates its effectiveness.
>
> > **Q2**: From my perspective, the proposed PSER is related to PPO and DPO methods in terms of regularization. It would be great there was similar theoretical justification (formally or informally) for this proposed loss term.
>
> **A2**: We attempt to explain the relationship between the PPO algorithm and our method from the gradient perspective. Both the PPO method and our method impose additional constraints on the learning objective. Theoretically, the PPO algorithm limits the gradient magnitude of the policy by maintaining a bounded update on the policy ratio. Our method introduces an additional constraint term $R$ in the loss function to smooth the gradient, by using the refined target label generated from the policy itself as a constraint. Although the forms of the constraints differ, both aim to smooth the gradients, thus ensuring the stability of model training and enhancing model robustness. DPO optimizes the policy directly based on preference information (e.g., human choices) rather than using the old policy to stabilize training, which is intuitively and theoretically different from PPO and the proposed method.
>
> > **Q3**: I do not see tendency in Fig. 3. Is there results with very few context lengths? eg. {1, 5, 10}
>
> **A3**: We run the experiments with the setting of shot context lengths. As shown in Table T3 in the uploaded pdf in the "global" response, the performances of DT and DM both drop substantially due to the lack of the contextual information. However, the performance degradation of DM is significantly less than that of DT. This suggests that our proposed DM effectively preserves the information in the input sequences.

---

> ### Author Response · Authors · 2024-08-12
> **Kind reminder**
>
> Thank Reviewer qjYZ again for the valuable comments. We have provided the response to each of the concerns raised in the review,  and we are eager to continue the conversation. As the interactive discussion window will close soon, we kindly invite the reviewer to read our response to see if there are any further questions.

---

### Official Review · Reviewer_vt4f · 2024-06-28

**Soundness:** 3
**Presentation:** 3
**Contribution:** 2
**Rating:** 6
**Confidence:** 3

**Summary:**

This paper tackles the sequential decision-making problem in an offline RL setting. The authors propose Decision Mamba (DM), an extension of Mamba to adapt to the problem. There are 3 main technical contributions: (a) DM architecture (a mix of fine-grained and coarse-grained SSM modules), (b) progressive self-evolution regularization (PSER) for label robustness, and (c) self-supervised loss (predicting states and rewards as well as action labels). The experiments show that DM outperforms baselines, including transformer-based approaches.

**Strengths:**

1. Applying Mamba-like models to the offline RL problem sounds natural and an interesting direction.
1. Empirical results look nice. DM significantly beats baselines.
1. The paper is clearly written and easy to follow.

**Weaknesses:**

1. It is somewhat unclear why that design choice on DM was made.
1. The technical contributions (a,b,c) would be independent and not tightly connected. I mean, (b) and (c) are possibly also effective to other methods like decision transformer.

**Questions:**

Regarding fine-grained and coarse-grained modeling, h^FG contains a conv layer, but h^CG doesn't. Why does the difference come out? What's the motivation?

Are PSER and self-supervised loss also effective against baselines such as decision transformers? Have you conducted any experiments on that?

**Limitations:**

Limitations are addressed.

---

> ### Author Rebuttal · Authors · 2024-08-07
>
> We thank reviewer vt4f for the time in evaluating our work. We now answer the concerns raised in the comments below.
>
> > **Q1**: It is somewhat unclear why that design choice on DM was made.
>
> **A1**: Compared to the transformer architecture, the state space model (used in mamba) has advantages in capturing historical information when modeling a sequence [1,2,3]. This is further confirmed by the supplementary experimental results in Table T5 in the uploaded pdf in the "global" response, where the mamba architecture (i.e., mamba_original) surpasses DT significantly. Furthermore, we made a modification to the mamba architecture specifically for Offline RL tasks by proposing the fine-grained SSM module. As shown in Table 3 in the submission, it brings an unignorable improvement (from 79.2 to 83.2). In addition, we further compare the effects of the proposed learning strategies, i.e., PSER and ILO, on mamba and transformer architectures. As shown in **the answer to the common concern in the "global" response**, although PSER and ILO improve the performance of DT substantially, DM still benefits more from these two strategies as it outperforms *DT w/ PSER&ILO* by a large margin. Thus, we make these design choices on DM.
>
> > **Q2**: The technical contributions (a,b,c) would be independent and not tightly connected. I mean, (b) and (c) are possibly also effective to other methods like decision transformer.
>
> **A2**: Please refer to **the common concern in the "global" response** for the details, where we provide more experimental results of the proposed learning strategies, i.e., PSER and ILO, on other methods.
>
> > **Q3**: Regarding fine-grained and coarse-grained modeling, $h^{\mathrm{FG}}$ contains a conv layer, but $h^{\mathrm{CG}}$ doesn't. Why does the difference come out? What's the motivation?
>
> **A3**: Intra-step relationships among the **S**tate, **A**ction, and **R**eturn-to-go (**SAR**) is one kind of the fine-grained information. It is well known that convolution neural networks (CNNs) are good at extracting local fine-grained features via small filters and sliding windows. Thus, we use the convolution layer to obtain the fine-grained features $h^{\mathrm{FG}}$, modeling the dependencies among each intra-step SAR. Regaring coarse-grained modeling, we use the state space model (gate-based architecture) directly to model the whole sequence to obtain coarse-grained features $h^{\mathrm{CG}}$. The fine-grained features  $h^{\mathrm{FG}}$ are then fused with the coarse-grained features $h^{\mathrm{CG}}$ in each Decision Mamba block, thus obtaining the multi-grained features, as shown in Figure 1 in the submission. We have also conducted experiments to validate the effectiveness of multi-grained information. As shown in Table 3 in the submission,  the performance of DM drops significantly without extracting multi-grained features (i.e., DM w/o GB), which demonstrates its effectiveness.
>
> > **Q4**: Are PSER and self-supervised loss also effective against baselines such as decision transformers? Have you conducted any experiments on that?
>
> **A4**: Please refer to **the common concern in the "global" response** for the details, where we provide more experimental results of the proposed learning strategies, i.e., PSER and ILO, on other methods.
>
> ## Reference
>
> [1] Gu A, Dao T. Mamba: Linear-time sequence modeling with selective state spaces. Arxiv 2023
> [2] Zhu L, Liao B, Zhang Q, et al. Vision mamba: Efficient visual representation learning with bidirectional state space model. ICML 2024.
> [3] Dao T, Gu A. Transformers are SSMs: Generalized models and efficient algorithms through structured state space duality. ICML 2024.

---

> ### Author Response · Authors · 2024-08-12
> **Kind reminder**
>
> Thank Reviewer vt4f again for the valuable comments. We have provided the response to each of the concerns raised in the review,  and we are eager to continue the conversation. As the interactive discussion window will close soon, we kindly invite the reviewer to read our response to see if there are any further questions.

---

> > ### Comment · Reviewer_vt4f · 2024-08-14
> > **To authors**
> >
> > Thank you for your response. My concerns have been addressed, and the additional experiments are convincing. I will raise my score.

---

> > > ### Author Response · Authors · 2024-08-14
> > >
> > > We sincerely thank Reviewer vt4f for the positive feedback and the decision to raise the score. We will make these points raised in the review more clear in the next version of this paper. Thank you again.

---

### Official Review · Reviewer_2Wf9 · 2024-07-10

**Soundness:** 3
**Presentation:** 3
**Contribution:** 3
**Rating:** 5
**Confidence:** 4

**Summary:**

The paper proposes a robust method based on Mamba for Offline Reinforcement Learning. Additionally, the paper using the knowledge of the past policy to refine the noisy labels as supervision avoids the model fitting the noisy trajectories. To better train the model, the paper introduce the inverse training goals which simultaneously predict action, state, and RTG for better robustness. Extensive experiments demonstrated the effectiveness of proposed model and training strategy.

**Strengths:**

1.	Introducing Mamba from the perspective of capturing historical information is interesting, and there are also some modifications to the Mamba architecture itself.

2.	Using easy-to-implement methods to refine noise labels using past policies to avoid overfitting onto suboptimal paths.

3.	A novel training method that enhances the robustness of the model by incorporating the next stage's state and RTG into the prediction.

**Weaknesses:**

1.	Lack of explanation for extracting intra step relationships: The paper lacks a specific explanation and verification for how intra-step relationships are extracted, as mentioned in the contributions.

2.	Lack of visual experiments on action changes affected by Formula16 and whether overfitting has truly been avoided: The paper only demonstrates the effectiveness of Formula 16 from the perspective of metrics. Can the effect of Formula 16 be visualized to prove that it preventS the model from overfitting to the suboptimal path and to show the impact it had on the actions compared with actions without Formula 16.

3.	Lack of specific experimental settings for the three hyperparameters in equation 23: The paper mentions improving the robustness of the model by changing the training objectives during the training phase, but there is a lack of experiments on the specific coefficients of the losses for the three parts and the impact of different hyperparameter values.

**Questions:**

1.	Have you considered applying the self-evolution regulation method and inverse training to other transformer-based methods to improve performance? Are these two methods universally applicable?

2.	The formula of refined target at k-th is too simple which is a linear formula. Can it be made more complex? Additionally, the parameters of linear formulas are generally learned through learning. Can they also be learned in your work.

3.	As mentioned in lines 183-187, there is a lack of explanation as to why Formula 16 does not introduce more historical information such as ak-2, ak-3, etc.

**Limitations:**

Despite not utilizing Mamba's computationally efficient capability to capture long-range dependencies, a window length was still set.

---

> ### Author Rebuttal · Authors · 2024-08-07
>
> We thank reviewer 2Wf9 for the efforts in reviewing our work. We have provided detailed explanations and additional experiments to address your concerns.
>
> > **Q1**: Lack of explanation for extracting intra step relationships...
>
> **A1**: Intra-step relationships mean the potential causal relationships among the **S**tate, **A**ction, and **R**eturn-to-go (**SAR**) in a single step. The intra-step relationship is one kind of the fine-grained information. It is well known that convolution neural networks (CNNs) are good at extracting local fine-grained features via small filters and sliding windows. Thus,  we use CNNs to model the  intra-step dependencies among SAR in each step. The extracted fine-grained features are then fused with the original coarse-grained features in each Decision Mamba block, as shown in Figure 1 in the submission.
>
> We have also conducted experiments to validate the effectiveness of intra-step relationships. As shown in Table 3 in submission,  the performance of DM drops significantly without intra-relationship (i.e., DM w/o GB), which demonstrates its effectiveness.
>
> > **Q2**: Lack of visual experiments on action changes affected by Formula16...
>
> **A2**: In addition to the quantitive metrics as shown in Table 3 in the paper, we actually have already provided the visualization of the action distribution in Figure 6 in Appendix E. We assume the action distribution learned from the medium-expert dataset as the ground truth. Even with high levels of noise in the training data (i.e., the medium-replay dataset), our DM still learns a action distribution highly consistent with the "ground truth". In contrast, the action distributions learned by DT are far away from the "ground truth". It demonstrates the effectiveness of Formula 16. In addition, we also provide an additional visualization comparing the agent's motion in the Walker-M-R task when trained with DT and DM as shown in the Figure F1 in the uploaded pdf.
>
> > **Q3**: Lack of specific experimental settings for the three hyperparameters...
>
> **A3**: $\lambda_1$ controls the contribution of the first term that learns the refined action target. As the action target is the primary goal for learning a policy, $\lambda_1$ is supposed to be set to a great value, i.e., >0.5. $\lambda_2$ and  $\lambda_3$ control the contributions of the losses of predicting  the next state and return-to-go (RTG), respectively. Since the state and RTG predictions are not the goal of the policy but serve as the regularizers, we assign $\lambda_2$ and $\lambda_3$ with small values. We further conduct experiments with different values for the three hyperparameters  $\lambda_1$,  $\lambda_2$, and $\lambda_3$, i.e., **S1**: $\{\lambda_1=1, \lambda_2=0, \lambda_3=0\}$; **S2**: $\{\lambda_1=0.9, \lambda_2=0.05, \lambda_3=0.05\}$ (used in our paper); **S3**: $\{\lambda_1=0.8, \lambda_2=0.1, \lambda_3=0.1\}$; and **S4**: $\{\lambda_1=0.6, \lambda_2=0.2, \lambda_3=0.2\}$.
>
> As shown in Table T2 in the uploaded pdf in the "general" response, the best performance is achieved with the **S2**. The performances of **S2** and **S3** are very close, which demonstrates DM is not sensitive to the hyperparameters. In addition, when $\lambda_1$ is reduced from 0.9 to 0.6, the performance (i.e., **S4**) drops significantly. The reason is that too small $\lambda_1$ leads to underfitting the action target. Furthermore, when the weights $\lambda_2$ and $\lambda_3$ for the regularizers are set to 0, the performance (i.e., **S1**) decreases by 1.4 compared to **S2**, which validates the effectiveness of these two regularizer terms.
>
> > **Q4**: Have you considered applying the self-evolution regulation method ...
>
> **A4**: Please refer to **the common concern in the "global" response** for the details, where we provide more experimental results of the proposed learning strategies, i.e., PSER and ILO, on other methods.
>
> > **Q5**: The formula of refined target at k-th is too simple...
>
> **A5**: $\beta_k$ controls the contribution of the knowledge from the past policy for refining the action label. As the training progresses, the refined action tends to be more accurate, and thus $\beta_k$ is supposed to be greater gradully. Therefore, $\beta$ can be predicted with the loss of fitting the action target.
>
> $\beta_k=\exp(-\lambda\cdot\mathcal{L}_{k})=\exp(-\lambda\cdot||\hat{a}_k-a_k||^2) $
>
> The experimental result is shown in Table T4 in the uploaded pdf, where DM-Learning denotes predicting $\beta_k$ using the above strategy, and DM-Linear denotes using the linear formula for $\beta_k$ as given in the paper. It can be observed that the results of the above two methods do not differ significantly. Therefore, we take the simple linear formula in this paper.
>
> > **Q6**: There is a lack of explanation as to why Formula 16 does not introduce more historical information.
>
> **A6**: Thanks for the good suggestion. We conduct additional experiments to investigate the effects of more historical information. Specifically, we define the $\tilde{a}_k=\beta {a}_k+(1-\beta)\tilde{a}\_{k-1}$ following an exponential moving average method, where k goes from 1 to the final step and $\beta$ is set to 0.9. As shown in Table T6 in the uploaded pdf in the "global" response, the performance experiences a slight decrease. We speculate that the knowledge from the policy at the early steps is not accurate, and thus introducing too much historical information may accumulate the errors, hurting its performance.
>
> > **Q7**: Despite not utilizing Mamba's... a window length was still set.
>
> **A7**:  For fair comparison, we follow the existing literature [8, 23, 42, 43, 58] to set the window length. Additionally, we conduct experiments with a larger context length. As shown in Figure 3 of the original submission and Table T3 in the uploaded pdf, the experimental results remain consistent that DM surpasses other transformer-based methods substantially.

---

> > ### Comment · Reviewer_2Wf9 · 2024-08-12
> > **Post-rebuttal comments**
> >
> > Thanks for the responses! I would like to keep the rating and looking forward to the improved revision.

---

> ### Author Response · Authors · 2024-08-12
>
> Thank you for your encouraging feedback and for recognizing our efforts to address your concerns.
>
> We are dedicated to continuously improving our work and would greatly appreciate any additional suggestions you may have to further enhance the quality of our work. If there are specific aspects where you believe further improvements could positively impact the score, we would be grateful for your advice.
>
> Thank you once again for your time and valuable insights.

---

### Official Review · Reviewer_K8tX · 2024-07-11

**Soundness:** 3
**Presentation:** 2
**Contribution:** 3
**Rating:** 6
**Confidence:** 4

**Summary:**

This paper introduces Decision Mamba, an offline RL backbone based on State Space Models. It enhances policy robustness by integrating a fine-grained SSM module alongside the original coarse-grained SSM in Mamba. Meanwhile, it adopts a progressive self-evolution regularization to prevent the policy from overfitting the noisy labels. Extensive experiments against a broad baseline demonstrate its effectiveness.

-----------Post Rebuttal-----------------

Main concerns resolved. Update rating from BA to WC.

**Strengths:**

- Mamba have shown good performance across NLP and CV tasks and applying Mamba to offline reinforcement learning is an interesting direction.
- The paper conducts a comprehensive comparison with SOTA offline RL methods.

**Weaknesses:**

- The advantages of DM appear somewhat constrained when considering variance. It appears that performance enhancements stem more from the PSER rather than the Mamba architecture. Combining PSER with other architectures, such as transformers, could potentially yield superior performance outcomes.
- In the experiments investigating various context lengths, the range examined is rather restricted, and there appears to be no notable difference in performance as the context length increases in either DT or DM. It may be beneficial to conduct experiments using longer context lengths to assess potential performance variations.
- It is not appropriate to assume that readers are very familiar with Mamba related terminology. Enhancing the methodology section with additional details to explain such terms would improve readability. For instance, the term "group branch" used in the ablation experiments lacks an earlier clear definition within the text.

**Questions:**

- Table 3 indicates a significant impact of PSER on performance. Given its compatibility with other proposed baselines, it would be worth testing whether adding that into them would improve their performance results.
- The context length in Mujoco environments is relatively short, minimizing the need for extensive historical temporal information. Can you provide experimental data on Atari environments to assess DM's capability in handling longer contextual sequences?
- More experiments on Multi-Grained SSM modules are requested to bolster the supporting evidence for their effectiveness.

**Limitations:**

Unable to surpass the performance of the behavior policy, the model performs poorly on a suboptimal dataset.

---

> ### Author Rebuttal · Authors · 2024-08-07
>
> We thank reviewer K8tX for the thoughtful comments. We now address the concerns below.
>
> > **Q1**: (1) The advantages of DM appear somewhat constrained when considering variance. (2) It appears that performance enhancements stem more from the PSER rather than the Mamba architecture. (3) Combining PSER with other architectures, such as transformers, could potentially yield superior performance outcomes.
>
> **A1**: (1) The performance variance mainly stems from the nature of the tasks in Mujoco. As shown in Table T1 in the uploaded pdf in the "global" response, the variance of each of the SOTA methods (including DT [1], EDT [2], and LaMo [3]) is quite large on the Mujoco tasks. The average variance of DT, EDT, LaMo, and DM are 1.9, 4.6, 2, and 2.2, respectively. It can be observed that DM has a similar variance comparing with other methods while its performance is substantially higher than them.
>
> (2) Although PSER improves the performance substantially, mamba also brings significant improvements. As shown in Table 1 and Table 3 in the submission, the performance of original DT is 75.8 while the performance of the proposed DM without PSER is 77.2, which shows an unignorable improvement (+1.4). When combing with PSER, the proposed DM is further strengthened.
>
> (3) Following the good suggestion, we conduct extra experiments combining PSER with BC and DT. Please refer to **the common concern in the "global" response** for the details. In summary, PSER improves the performances of DT and BC by 1.8 and 1.9, respectively. PSER is also beneficial to other methods, while the best performance is still achieved by the proposed Decision Mamba.
>
> > **Q2**: In the experiments investigating various context lengths, the range examined is rather restricted, and there appears to be no notable difference in performance as the context length increases in either DT or DM. It may be beneficial to conduct experiments using longer context lengths to assess potential performance variations.
>
> **A2**: We conduct additional experiments with longer context. As shown in Table T3 in the uploaded pdf in the "global" response, we first increase the context length to 200, both DM and DT experience significant performance degradation. When further enlarging the context length, the GPU (NVIDIA A800-80G) is out of memory. Then, we go to the opposite direction, trying extreme shorter context length, e.g., 5. The performance of DT and DM both drop substantially due to the lack of the contextual information. However, the performance degradation of DM is significantly less than that of DT. This suggests that our proposed DM effectively preserves the information in the input sequences.
>
> > **Q3**: It is not appropriate to assume that readers are very familiar with Mamba related terminology. Enhancing the methodology section with additional details to explain such terms would improve readability. For instance, the term "group branch" used in the ablation experiments lacks an earlier clear definition within the text.
>
> **A3**: Thanks for your good suggestion and we will add more background about mamba to the paper, e.g., supplementing the mamba background in the Related Work section 2.2 and the Method section 3.1. We will also add more explanations about the "group branch". The group branch denotes the combination of the coarse-grained module branch and the fine-grained module branch.
>
>
> > **Q4**: Table 3 indicates a significant impact of PSER on performance. Given its compatibility with other proposed baselines, it would be worth testing whether adding that into them would improve their performance results.
>
> **A4**: Please refer to **A1(3)** above.
>
> > **Q5**: The context length in Mujoco environments is relatively short, minimizing the need for extensive historical temporal information. Can you provide experimental data on Atari environments to assess DM's capability in handling longer contextual sequences?
>
> **A5**: Regarding the ability to handle longer context sequences, please refer to **A2**. The results show that DM has a significant advantage over DT under the settings of both long and short context. It is a good suggestion to extend experiments in the Atari environment. However, due to the limited rebuttal time, we are unable to conduct comprehensive experiments regarding various context length and methods in Atari. We will supplement this experiment in the future.
>
> > **Q6**: More experiments on Multi-Grained SSM modules are requested to bolster the supporting evidence for their effectiveness.
>
> **A6**: As shown in Table 3 in the submission, DM with Coarse-Grained SSM module (i.e., DM w/o GB) achieves the performance of 79.2, while DM with Multi-Grained SSM module (i.e., the proposed DM) obtains the performance of 83.2. It demonstrates the effectiveness of Multi-Grained SSM modules.
>
> ## Reference
>
> [1] Chen L, Lu K, Rajeswaran A, et al. Decision transformer: Reinforcement learning via sequence modeling. NeurIPS 2021.
> [2] Wu Y H, Wang X, Hamaya M. Elastic decision transformer. NeurIPS 2024.
> [3] Shi R, Liu Y, Ze Y, et al. Unleashing the power of pre-trained language models for offline reinforcement learning. ICLR 2024.

---

> > ### Comment · Reviewer_K8tX · 2024-08-12
> >
> > I thank the authors for providing detailed responses to my comments. Most of my concerns are solved after reading the rebuttal materials. I also read the comments from other reviewers and decide to update my rating from Boardline Accept to Weak Accept.
> >
> > Good luck to the authors for the final decision of this work.
> >
> > Best wishes,

---

> ### Author Response · Authors · 2024-08-12
>
> We greatly appreciate Reviewer K8tX for the insightful feedback and the decision to raise the score. In the next version, we will offer more background about mamba, and we will include additional experiments to further strengthen our work.

---

### Author Rebuttal · Authors · 2024-08-07

# "Global" Response
We thank all the Reviewers, ACs, SACs, and PCs for their efforts and valuable comments. In terms of the idea of this paper, all the reviewers have recognized that exploring mamba on the Offline RL is interesting, and we have also made some modifications to mamba specifically for Offline RL tasks. In terms of the experiments, almost all the reviewers (Reviewer K8tX, vt4f, qjYZ) have recognized that the proposed method beats the state-of-the-art baselines significantly. In summary, both the idea and the experiments have been acknowledged by almost all the reviewers. In this "global" response, we address the common concern raised by the reviewers, while we respond to the other concerns raised by each reviewer in separate rebuttals.

---

Since we improve the effectiveness on offline RL tasks from two novel perspectives, i.e., the model architecture (mamba) and the learning strategies (PSER and ILO), the reviewers have a common concern about **whether the proposed learning strategies, i.e., PSER and ILO, are beneficial to other methods**.

>  **Common Concern: Whether the proposed learning strategies, i.e., PSER and ILO, are beneficial to other methods.**

**Answer**: We conduct additional experiments to evaluate **the effects of the proposed learning strategies (PSER and ILO) on other methods**. Specifically, we apply PSER and/or ILO to two classic Offline RL methods, i.e., Decision Transformer (DT) and Behavior Cloning (BC). As shown in the following table, PSER improves the performances of DT and BC by 1.8 and 1.9 on average, respectively. ILO improves the performance of DT by 0.9 (ILO is not applicable to BC). When combining PSER and ILO, the improvement on DT is further strengthened (+4.2). Therefore, the proposed learning strategies are also beneficial to other methods. However, the best performance is still achieved by the proposed DM, which demonstrates the superiority of DM.

|               | Halfcheetah-M | Hopper-M | Walker-M | Halfcheetah-M-E | Hopper-M-E | Walker-M-E | Halfcheetah-M-R | Hopper-M-R | Walker-M-R | Avg  |
| ------------- | ------------- | -------- | -------- | --------------- | ---------- | ---------- | --------------- | ---------- | ---------- | ---- |
| **DT**        | 42.6          | 70.4     | 74.0     | 87.3            | 106.5      | 109.2      | 37.4            | 82.7       | 66.2       | 75.8 |
| w/ PSER       | 43.1          | 69.3     | 79.8     | 90.4            | 109.3      | 110.3      | 39.3            | 81.3       | 75.6       | 77.6 |
| w/ ILO        | 43.0          | 72.2     | 77.2     | 90.6            | 108.3      | 109.7      | 39.6            | 78.4       | 71.5       | 76.7 |
| w/ PSER & ILO | 43.4          | 83.9     | 82.4     | 92.7            | 111.2      | 110.3      | 41.5            | 79.7       | 75.6       | 80.0 |
| **BC**        | 42.2          | 55.6     | 71.9     | 41.8            | 86.4       | 80.2       | 2.2             | 23.0       | 47.0       | 50.0 |
| w/ PSER       | 43.0          | 57.3     | 75.1     | 42.9            | 88.7       | 83.2       | 2.2             | 28.5       | 46.9       | 51.9 |
| w/ ILO        | N/A           | N/A      | N/A      | N/A             | N/A        | N/A        | N/A             | N/A        | N/A        | N/A  |
| **DM**        | 43.8          | 98.5     | 80.3     | 93.5            | 111.9      | 111.6      | 40.8            | 89.1       | 79.3       | 83.2 |

---

### Author Response · Authors · 2024-08-13
**Kind reminder**

Dear Reviewers, Area Chairs, Senior Area Chairs, and Program Chairs,

We are grateful to all the reviewers for the insightful comments and constructive suggestions and to the area chairs, senior area chairs and program chairs for the great service to the community. We have received the responses of Reviewer K8tX and 2Wf9, and hope our responses have adequately addressed the concerns of Reviewer vt4f and qjYZ. Please let us know if you have further questions. We take this as a great opportunity to improve our work and shall be grateful for any additional feedback you could give us.

Thanks,

Authors of Submission#13292

---

### Public Comment · ~Prabhant_Singh2 · 2024-11-29
**Code not available on URL**

Dear authors the link you provided is not available on the GitHub repo. Can you please correct that?

---

> ### Public Comment · ~Qi_Lv1 · 2024-11-29
>
> Thank you for your interest. We are currently organizing the code and will release the complete version shortly in the repository linked in our paper.

---

### Decision · Program_Chairs · 2024-09-25

**Decision:**

Accept (poster)

**Comment:**

This paper proposes "Decision Mamba," a novel multi-grained state space model (SSM) with self-evolution regularization tailored for offline reinforcement learning (RL) tasks. The paper addresses three key challenges in offline RL: (1) insufficient utilization of historical temporal information, (2) overlooking intra-step relationships, and (3) overfitting suboptimal trajectories. The proposed method demonstrates robust performance improvements across various tasks by integrating fine-grained and coarse-grained SSM modules and employing a self-evolving policy learning strategy.
This paper received all positive recommendations (three weak accepts and one borderline accept). The major concern from reviewers was about the learning strategies, which was addressed in the rebuttal. The AC agrees with the reviewers’ assessments and recommends accepting the paper.